# Morphology Control and Metallization of Porous Polymers Synthesized by Michael Addition Reactions of a Multi-Functional Acrylamide with a Diamine

**DOI:** 10.3390/ma14040800

**Published:** 2021-02-09

**Authors:** Naofumi Naga, Minako Ito, Aya Mezaki, Hao-Chun Tang, Tso-Fu Mark Chang, Masato Sone, Hassan Nageh, Tamaki Nakano

**Affiliations:** 1Department of Applied Chemistry, College of Engineering, Shibaura Institute of Technology, 3-7-5 Toyosu, Koto-ku, Tokyo 135-8548, Japan; ad14094@shibaura-it.ac.jp; 2Graduate School of Engineering and Science, Shibaura Institute of Technology, 3-7-5 Toyosu, Koto-ku, Tokyo 135-8548, Japan; mc19002@shibaura-it.ac.jp; 3Institute of Innovative Research, Tokyo Institute of Technology, Yokohama, Kanagawa 226-8503, Japan; hct2498@gmail.com (H.-C.T.); chang.m.aa@m.titech.ac.jp (T.-F.M.C.); sone.m.aa@m.titech.ac.jp (M.S.); 4Institute for Catalysis and Graduate, School of Chemical Sciences and Engineering, Hokkaido University, N 21, W 10, Kita-ku, Sapporo 001-0021, Japan; science_as2000@yahoo.com (H.N.); tamaki.nakano@cat.hokudai.ac.jp (T.N.); 5Integrated Research Consortium on Chemical Sciences, Institute for Catalysis, Hokkaido University, N 21, W 10, Kita-ku, Sapporo 001-0021, Japan

**Keywords:** porous polymer, Michael addition reaction, acryl amide, diamine, electroless plating

## Abstract

Porous polymers have been synthesized by an aza-Michael addition reaction of a multi-functional acrylamide, *N*,*N*′,*N*″,*N*‴-tetraacryloyltriethylenetetramine (AM4), and hexamethylene diamine (HDA) in H_2_O without catalyst. Reaction conditions, such as monomer concentration and reaction temperature, affected the morphology of the resulting porous structures. Connected spheres, co-continuous monolithic structures and/or isolated holes were observed on the surface of the porous polymers. These structures were formed by polymerization-induced phase separation via spinodal decomposition or highly internal phase separation. The obtained porous polymers were soft and flexible and not breakable by compression. The porous polymers adsorbed various solvents. An AM4-HDA porous polymer could be plated by Ni using an electroless plating process via catalyzation by palladium (II) acetylacetonate following reduction of Ni ions in a plating solution. The intermediate Pd-catalyzed porous polymer promoted the Suzuki-Miyaura cross coupling reaction of 4-bromoanisole and phenylboronic acid.

## 1. Introduction

Various types of porous polymers have been prepared by polymerization-induced phase separation (vinyl type monomers [1,2,3,4,5,6,7,8,9,10,11,12,13,14,15,16,17,18,19,20,21,22], click type polymerizations such as epoxy-amine reaction [23,24,25,26,27,28,29,30,31,32,33,34], epoxy-thiol reaction [35,36,37], thiol-ene/yene [38,39,40,41,42,43,44], thiol-(meth) acrylate [45,46,47]) and temperature induced phase transfer [48,49,50,51,52,53,54,55,56,57,58,59,60,61,62,63,64,65,66,67,68]. Applications of the porous polymers have been developing in various fields, for examples separation columns for liquid chromatography [1,2,4,5,6,7,8,10,11,12,13,14,15,16,17,18,24,25,27,28,29,30,32,35,36,37,39,41,42,43,45,46,50], catalyst supports [20,68,69,70], conductive material supports [71,72], battery separators [34], binders of metal and plastic adhesion [73], cell cultivation scaffolds [47,74,75,76], and so on. The porous structure and properties, such as mechanical properties (hard or soft), thermal properties, stability (stable or degradable), affinity with solvents (hydrophilic or hydrophobic), of the porous polymers can be widely controllable by molecular (chemical) structure and formation process (reactions conditions) the polymers. Variations of the structure and properties of the porous polymers should be desirable for expanding their applications.

We have been developing several types of porous polymers synthesized by addition reactions between a multi-functional monomer, as a source of joint parts (“joint-source monomer”), and a α,ω-bifunctional monomer, as a source of linker parts (“linker-source monomer)”, in some solvents via polymerization-induced phase separation. In our previous studies, combinations of multi-functional thiol, polyol, acrylate, and amine as the joint-source monomer and diisocyanate, divinyl ether, dithiol, diamine and diacrylate compounds as the linker-source monomer could yield porous polymers under specific reaction conditions [77,78,79,80,81,82,83]. The porous polymers showed surface morphology formed by co-continuous monolithic structures or connected spheres. These structures should be generated by phase separation via spinodal decomposition. Addition of a pore-generator (porogen) is useful to accelerate the phase separation. Polymeric compounds or surfactants are commonly used as the porogen. However, these porogenic materials must be removed from the obtained porous polymers before use. The joint-linker concept makes it possible to obtain porous polymers without a porogen. Purification of the resultant porous polymers is possible by washing with a solvent and subsequent drying. Combinations of the joint-source and linker-source monomers should expand the variety of the porous polymers with various chemical structures, properties and functions.

We previously prepared some porous polymers containing hydrolytically degradable groups in their polymer networks. For examples, the addition reaction of triphenol and poly(ethylene glycol)divinyl ether compounds in acetonitrile in the presence of an acid catalyst yielded a porous polymer accompanied by formation of acetal group in the polymer network [83]. The porous polymer was easily hydrolytically degradable under atmospheric conditions. Some porous polymers have been obtained from Michael addition reactions between multi-functional thiol or amine compounds and poly(ethylene glycol)diacrylate. The resultant porous polymers were quickly hydrolytically degradable in acidic or basic water. The porous polymers were also gradually degraded by hydrolysis of acrylate groups in the polymer networks caused by atmospheric moisture [77,80]. Degradability is a useful feature for products which need to display a low environment load and/or biodegradability. By contrast, it is not suitable for the applications which need long lifetimes, such as columns or filters for separation, battery separators, supports for catalysts and so on. The hydrolytic stability of the porous polymer should also make it possible to handle the polymer in H_2_O. This feature would be useful to modify surface of the polymer by various chemical reactions under acidic or basic conditions.

The three-dimensional porous structure strongly affects the performance of porous polymers in their applications. A co-continuous monolithic structure, which is formed by the coexistence of a polymer backbone and vacant spaces, is superior for solvents to flow through the porous polymer under low pressure. The porous polymers with co-continuous monolithic structures can be prepared by polymerization-induced phase separation via spinodal decomposition. Some reaction systems based on the joint and linker concept successfully yielded porous polymers formed by a co-continuous monolithic structure [77,80,81,82,83]. However, the reaction systems and reaction conditions which form the desirable porous structures are limited. Highly internal phase emulsion (HIPE) is another effective way to prepare porous polymers with co-continuous structures, which are composed by connected holes [84]. HIPE is applicable in the reaction systems using H_2_O as the solvent with an emulsifier. If H_2_O can be used as a solvent in the polymerizations based on the joint and linker concept, HIPE should be usable to form porous polymers with controlled surface morphology.

As the next step of our development, we planned to synthesize hydrolytic stable porous polymers based on the joint and linker concept in H_2_O. We came to an idea to use a monomer with an acrylamide group for the polymerizations. The acrylamide group shows hydrolytic stability in H_2_O and high reactivity with amines via the aza-Michael addition reaction. We select a multi-functional acrylamide compound, as the joint-source monomer. We report herein the synthesis of porous polymers by the aza-Michael addition reaction of commercial available compounds *N,N′,N″,N‴*-tetraacryloyltriethylenetetramine (AM4) and hexamethylene diamine (HDA) in H_2_O (Scheme 1). Our studies focus on effect of reaction conditions, monomer concentration and reaction temperature, on the surface morphology of the resulting porous polymers. The present reaction system produced porous polymers induced by phase separation via not only spinodal decomposition but also HIPE, depending on the polymerization conditions. In other word, the porous structure of AM4-HDA network polymers can be widely controlled by two types of phase separations without further addition of an emulsifier. Basic properties and some applications, plating, support for a catalyst, of the porous polymers were also investigated.

## 2. Materials and Methods

### 2.1. Materials

*N*,*N*′,*N*″,*N*‴-Tetraacryloyltriethylenetetramine (AM4) was kindly donated by Fujifilm Corporation (Tokyo, Japan), and used as received. 1,6-Hexanediamine (HDA, Fujifilm Wako Pure Chemical Industries, Osaka, Japan) was commercially obtained, and used as received.

SnCl_2_ (Tokyo Chemical Industry Co. Ltd., Tokyo, Japan), PdCl_2_ (Fujifilm Wako Pure Chemical Industries), and hydrochloric acid (Kanto Chemical Co., Inc., Tokyo, Japan) were commercially obtained and used as received. Ni-P electrolyte, which was consisted of nickel chloride (1.8 wt %), sodium hypophosphite (2.4 wt %), complexing agent (2.4 wt %), and ion-exchanged water (93.4 wt %), was commercially obtained from Okuno Chemical Industries Co., Ltd. (Osaka, Japan). Palladium(II) acetylacetonate (Pd(acac)_2_, Tokyo Chemical Industry Co. Ltd.), CO_2_ (99.99%, Nippon Tansan Co., Ltd., Tokyo, Japan) and polyoxyethylene lauryl ether (Tokyo Chemical Industry Co. Ltd.) were was commercially obtained.

Ethanol (EtOH, Kanto Chemical Co., Inc., Tokyo, Japan), K_2_CO_3_ (Kanto Chemical Co., Inc.), phenylboronic acid (Tokyo Chemical Industry Co. Ltd.), and 4-brompanisol (Tokyo Chemical Industry Co. Ltd.) were commercially obtained and used as received.

### 2.2. Synthesis of Porous Polymers

HDA has two NH_2_ groups. The reactions of AM4 and HDA were conducted considering HDA as bi-functional molecule (Case I, [acrylamide] = [NH_2_], feed molar ratio of AM4/HDA: 1/2) or tetra-functional molecule (Case II, [acrylamide] = [active hydrogen], feed molar ratio of AM4/HDA: 1/1), as shown in Scheme 1.

A reaction of AM4 with HDA (monomer concentration: 10 wt %, feed molar ratio of AM4/HDA: 1/2) is described as an example. AM4 (0.182 g, 0.51 mmol), H_2_O (2.70 mL), and HDA (0.117 g, 1.01 mmol) were added to a 20 mL of vial, and stirred by a vortex mixer for several minutes to make a homogeneous solution. The reaction solution was introduced to a 10 mL ampoule. After the ampoule was sealed, the reaction was conducted at the desired temperature for 24 h. The obtained porous polymer was washed with an excess of methanol with ultra-sonification for several hours. The porous polymer, which was obtained as a block, was air-dried at room temperature, and further dried in vacuo at 30 °C for 6 h. Code of this sample is defined as follows: I-20w-30d (case of AM4/HDA feed ratio-monomer concentration-temperature). The reactions with different AM4/HDA feed ratios and/or monomer concentrations were conducted using the same procedures.

### 2.3. Plating of Porous Polymer

#### 2.3.1. Method 1

AM4-HDA porous polymer (1.7 g) was immersed in H_2_O (10 mL) for 24 h. The wet porous polymer was soaked in a PdCl_2_/SnCl_2_ solution (50 mL) and transferred to a commercial available acidic Ni-P electrolyte “Top Nikoron VS” (Okuno Chemical Industries Co., Ltd.) at 70 °C for the Ni plating [85,86]. The plating was conducted for 5, 10, or 20 min.

#### 2.3.2. Method 2

AM4-HDA porous polymer (1.7 g), Pd(acac)_2_ (30.0 mg), and CO_2_ were introduced into a 50 mL high pressure reaction cell of the high-pressure experimental apparatus (JASCO Co., Ltd., Tokyo, Japan) at 70 °C and 15 MPa (supercritical carbon dioxide, sc-CO_2_) with agitation for 2 h for catalyzation. The Ni plating of the catalyzed porous polymer was conducted in the same cell in the presence of Ni-P electrolyte (37. 6 μL) and polyoxyethylene lauryl ether, C_12_H_25_(OCH_2_CH_2_)_15_OH, (200 μL) in sc-CO_2_ at 70 °C and 15 MPa with agitation for 30 min.

### 2.4. Suzuki-Miyaura cross Coupling Reaction with Pd Catalyzed Porous Polymer

Water (15 mL), EtOH (15 mL), K_2_CO_3_ (0.97 g, 7.0 mmol), phenylboronic acid (0.64 g, 5.25 mmol), and 4-brompanisol (0.65 g, 3.5 mmol) were added to a flask with a stirrer piece. A Pd(acac)_2_ catalyzed AM4-HDA porous polymer (0.44 g, Pd 2.18 wt %) was cut to small blocks about 1 mm, then was added into the flask and stirred at 55 °C for 3 h. The reaction solution was extracted with 100 mL of CHCl_3_, and collected oil phase was evaporated and dried at room temperature in vacuo for 6 h.

### 2.5. Analytical Procedures

The mechanical properties of the gels were investigated by the compression test with a Tensilon RTE-1210 apparatus (ORIENTEC Co. Ltd., Tokyo, Japan). The test samples were cut to 1 cm cube, and pressed at a rate of 0.5 mm/min at room temperature.

Scanning electron microscopy (SEM) images of the porous polymers or SEM/energy dispersive X-ray spectroscopy of Pd catalyzed or Ni plated AM4-HDA porous polymers were acquired by a JEOL JSM-7610F microscope with a LEI detector at an acceleration voltage of 3.0 kV or 20 kV, respectively. As prepared samples (without coating or treatment) were used for the observations. Ni content was determined by ZAF correlation method. Size distribution of spheres and holes in SEM images was evaluated by image analysis using a software of Image-J.

The surface area of the porous polymer was measured by nitrogen sorption using an Autosorb 6AG (Quantachrome Instruments, Boynton Beach, FL, USA), and determined by Brunauer-Emmett-Teller (BET) equation.

Thermal analysis of a polymer was conducted with a differential scanning calorimetry (DSC) of a DSC 8230 system (Rigaku, Tokyo, Japan). The sample was heated from room temperature to 400 °C at a rate of 10 °C/min under nitrogen atmosphere.

^1^H-NMR spectra were recorded on a JNM-LA300 spectrometer (JEOL, Tokyo, Japan) in pulse Fourier transform mode. The pulse angle was 45° and 32 scans were accumulated in 7 s of the pulse repetition. Dimethylsulfoxide-D_6_ was used as the solvent.

## 3. Results and Discussion

### 3.1. Synthesis and Structure of Porous Polymers

Aza-Michael addition reactions of AM4 and HDA in H_2_O were conducted under the conditions described in the experimental part. The reaction yielded porous polymers over a wide range of temperatures. Figure 1 shows production diagram of the AM4-HDA reaction system of Case I. The reaction systems with low monomer concentrations, less than 20 wt %, preferentially yielded porous polymers as blocks of ample shape. By contrast, the reaction systems with high monomer concentrations, 20 and 30 wt %, tended to yield gels. The increase of monomer concentration in the reaction system would convert the state from separated double phase to homogeneous single phase during the network formation. The reactions at high temperatures, 80–90 °C, yielded the porous polymers even in the reaction systems with high monomer concentration. In general, increase of the reaction temperature increases the solubility of the resultant polymer network, and would tend to form the gel with homogeneous phase. We shall return to this point of the unexpected result later.

Related surface area of an AM4-HDA porous polymer I-10w-25d evaluated by nitrogen adsorption/desorption (Appendix A) was 3.8 m^2^/g. The related surface areas of other porous polymers were too low to evaluate quantitatively. The low value of surface area is not derived from a “microporous” but rather a “macroporous” structure. The surface structure of the porous polymers was observed by SEM. Figure 2 shows SEM images of some AM4-HDA porous polymers prepared under the various conditions with Case I, feed molar ratio of AM4/HDA: 1/2 (histograms of diameter size distribution of spheres and/or holes are available in Appendix A). The porous polymer of I-20w-70d showed a surface morphology formed by connected spheres ranged from 1.8 to 5.2 μm diameter, as shown in Figure 2a and Appendix A. The corresponding porous polymers obtained at 80 °C or 90 °C, I-20w-80d or I-20w-90d, also showed a similar surface morphology (Figure 2b,c), and the diameter of the spheres (Appendix A) were larger than those of I-20w-70d. The surface morphology of these porous polymers would be induced by phase separation via spinodal decomposition. The spheres observed in SEM images would be fixed at the late stage of the spinodal decomposition, as illustrated in Scheme 2a. A rise of the reaction temperature should increase the polymerization rate and increase the solubility of the intermediate polymer network, which fixed the porous structure with large spheres at a later stage of the phase separation. The reaction with higher monomer concentration (25 wt %) at 80 °C, I-25w-80d, yielded a porous polymer, whose surface morphology was formed by connected spheres of 1.4–5.0 μm diameter and isolated holes of 1.1–5.9 μm diameter, as shown in Figure 2d and Appendix A. The corresponding reaction (25 wt %) at 90 °C, I-25w-90d, yielded a porous polymer whose surface morphology was formed by connected holes of 2.3–21.0 μm diameter, as shown in Figure 2e and Appendix A. A small portion of connected spheres was also observed in this porous polymer. An increase of the monomer concentration to 30 wt %, I-30w-90d, widened the diameter range of the holes (2.1–52.0 μm) in the resultant porous polymers, as shown in Figure 2f and Appendix A. One explanation for these results is that the intermediate AM4-HDA polymer network would behave like an emulsifier, and the holes in the porous polymers should be derived from highly internal phase emulsion (HIPE) [75], as illustrated in Scheme 2b. The size of holes should be derived from the size of aqueous droplets of the emulsion in the reaction system. The increase of monomer concentration and reaction temperature should increase the size of aqueous droplets caused by collapse (flocculation and/or coalescence) of the emulsion. The diameter size distribution of holes or co-existence of holes and spheres can be explained by inhomogeneity of the phase separations in those reactions. The monomer concentration (non-reacted monomers) in the reaction solutions decreased with progressing of the polymerization, and small holes and/or spheres would be fixed at the late stage of the polymerization. Another possibility is the co-existence of different monomer concentrations in the reaction system derived from phase equilibrium.

The reactions with Case II, feed molar ratio of AM4/HDA: 1/1, were conducted regarding HDA as a tetrafunctionalized monomer. The reactions also produced porous polymers. SEM images of some samples obtained in the reactions are shown in Figure 3. The reaction with 25 wt % of monomer concentration at 40 °C yielded the porous polymer II-25w-40d, whose surface morphology was formed by small spheres and isolated holes of less than 2 μm in diameter, as shown in Figure 3a. These structures should be derived from phase separation via spinodal decomposition (Scheme 2a) and HIPE (Scheme 2b), respectively. The reactions at higher temperatures, 60 °C and 80 °C, II-25w-60d and II-25w-80d, yielded porous polymers formed by isolated holes with large (more than 10 μm) and small (less than 5 μm) diameters, which would be induced by HIPE, as shown in Figure 3b,c and Appendix A. These results indicate that the reactions with Case II would induce HIPE even at low reaction temperatures due to the relatively high concentration of AM4 in these reaction conditions in comparison with the reactions with Case I. Increase of the monomer concentration (30 wt %) in the reaction system at 40 °C turned the morphology of the porous polymer into distorted connected holes (II-30w-40d), as shown in Figure 3d. The reaction at low temperature should decrease the solubility of the intermediated polymer network and induce phase separation via spinodal decomposition. Furthermore, the reaction with higher monomer concentration increased the rate of network formation. These factors should fix the porous structure at earlier stage of the phase separation with a microscopic co-continuous structure.

### 3.2. Properties of AM4-HDA Porous Polymer

#### 3.2.1. Mechanical and Thermal Properties

The mechanical properties of the AM4-HDA porous polymers, which were obtained in the reactions at 90 °C, were investigated by compression tests. All the porous polymers were soft and flexible and were not breakable under a compression of 50 N. Stress-strain curves of the AM4-HDA porous polymers obtained from the reaction systems with Case I, feed molar ratio of AM4/HDA: 1/2, are shown in Figure 4, and the results are summarized in Table 1. The Young’s modulus of the porous polymer increased with the increase in the monomer concentration of the reaction systems. An increase of the monomer concentration increased the filling space, which should increase the Young’s modulus of the porous polymers. The holes’ morphology formed in the reaction systems with high monomer concentration also induced higher Young’s modulus than the spheres’ morphology formed in the reaction systems with low monomer concentration (Figure 2), as previously reported in other reaction systems [81]. AM4-HDA porous polymer of Case II, II-30w-90d, showed higher Young’s modulus (Appendix A) than the corresponding porous polymer of Case I, I-30w-90d. The higher crosslinking density derived from higher acryloyl concentration (2.5 mmol) in the reaction system of Case II in comparison with corresponding reaction system with Case I (2.0 mmol) should induce higher Young’s modulus, as summarized in Table 1.

The thermal properties of an AM4-HDA porous polymer I-10w-25d were studied with DSC measurements. The sample was heated from room temperature to 400 °C at a rate of 10 °C/min under a nitrogen atmosphere (Appendix A). Endothermic peaks were detected at around 90 °C and 350 °C in the DSC profile. The endothermic peak at around 90 °C in the DSC profile should derive from the evaporation of water. The latter peak should be caused by thermal degradation of acrylamide groups in the polymer network. The porous polymer was heated on a hot plate from room temperature to 200 °C to observe the form. The porous polymer softened by heating, but the form did not change.

#### 3.2.2. Absorption of Solvents

The AM4-HDA porous polymer absorbed various solvents. Figure 5 shows the absorption capacity of a AM4-HDA porous polymer I-25w-10d (feed molar ratio of AM4/HDA: 1/2, monomer concentration in reaction solution: 10 wt %, preparation temperature 25 °C). The polymer absorbed 165–580 wt % of solvents based on the original weight. A linear relationship was observed between the density of most of the solvents and the weight gain (Appendix A), except for H_2_O and EtOH. The absorption capacities of H_2_O and EtOH (especially EtOH) were higher than those of other solvents based on their density. The calculated solubility parameter (SP) value (Fedors’ method [87]) of repeating unit of the AM4-HDA polymer network is 11.4 (cal/cm^3^)^1/2^. The SP value of H_2_O or EtOH is 23.4 (cal/cm^3^)^1/2^ or 12.7 (cal/cm^3^)^1/2^), respectively. The SP value of EtOH is close to that of the AM4-HDA repeating unit. The high absorption capacity of EtOH should be derived from the high affinity between the polymer network and EtOH. The absorption capacity of DMSO was lower than that of H_2_O, despite a closer SP value, 14.5 (cal/cm^3^)^1/2^, to the repeating unit in the network. Calculation of SP by Fedors’ method is normally applicable in solution systems without electrostatic interactions and/or dipolar interactions in cohesion between solute and solvent. One explanation of the present results is that hydrogen bonding between the AM4-HDA polymer network and hydroxyl group of H_2_O or EtOH should increase the affinity between the network and the solvent.

### 3.3. Plating of AM4-HDA Porous Polymer

Ni plating of AM4-HDA porous polymer was conducted after Pd catalyzation by Method 1, via catalysis in a PdCl_2_/SnCl_2_ solution, as described in the experimental section. Hydrogen bubbles were detected during the electroless plating reaction process. The plating turned the surface of the porous polymer black (Appendix A). These results mean that a Ni-P deposition reaction occurred on the porous structures. SEM images of the Ni plated porous polymer are shown in Figure 6. After the plating for 5 min, the metallization occurred among the polymer spheres, as shown in Figure 6b. Pd could deposit easily in the gap between the spheres at this state. Fusion of metallized polymer spheres by lateral growth, which is the typical growth of electroless plating on initial stage, was observed after the plating for 10 min, as shown in Figure 6c. The plating for 20 min induced rough surface due to excess plating of Ni on the Ni plated surface of Figure 6c by normal nodule growth, which is the typical phenomenon observed on second stage of electroless plating. Although these results showed possibility of Ni plating of the AM4-HDA porous polymer using Method 1, inside of the porous polymer was not plated by Ni. One explanation of the result is that the normal growth of Ni closed the pour structure on the surface by Ni.

Ni plating of AM4-HDA porous polymer was also conducted by Method 2, via catalyzation by Pd(acac)_2_ in sc-CO_2_ following plating, as described in the experimental procedures. Figure 7a,b show SEM images and Pd elemental mapping of the Pd catalyzed porous polymer, before plating. The element mapping showed the existence of elemental Pd over the surface of the porous polymer and Method 2 was effective to impregnate Pd into the porous structure. SEM image and Ni element mapping of the porous polymer after plating are shown in Figure 7c,d. In comparison with the results obtained by Method 1, Figure 6, the characteristic feature of the SEM image, Figure 7c, is a deep and complicated pore structure that remained on the surface after plating by Ni. Elemental Ni was detected on the whole surface of the porous polymer by element mapping, as shown in Figure 7d. The content of Ni was quite high (73.4 wt %). These experimental results by Method 2 offer these interesting insights: (1) Pd catalysts were deposited over all of the surface of the microporous structure; (2) Ni grew laterally on the Pd-catalyzed pore surface inside; (3) after covering the surface by Ni, further Ni growth occurred normally and the metallized polymer spheres were fused. This kind of metal growth in electroless plating was also found in Pd electroless plating on γ-alumina [88]. As widely accepted, metallization of complicated polymer structures is difficult. Although the present study is a typical example of metallization of Ni on complicated pore structure of polymeric material using sc-CO_2_, the precise control of the reaction conditions of the electroless plating would enable complete metal coverage of the microporous structures.

### 3.4. Coupling Reaction by Pd Catalyzed Porous Polymer

Suzuki-Miyaura coupling reactions of 4-bromoanisole and phenylboronic acid, Scheme 3, were conducted using Pd-containing AM4-HDA porous polymer, catalyzed in sc-CO_2_ for 2 h (by Method 2 before Ni plating). The reaction conversion of 4-methoxy biphenyl in crude reaction mixtures, determined by ^1^H-NMR spectroscopy, was 82%, and the isolated yield was 40%. The conversion of the corresponding reaction with Pd(acac)_2_ in homogeneous system was more than 98%. The reaction did not progress further with the original AM4-HDA porous polymer. Although the conversion was lower than with the homogeneous reaction system with Pd(acac)_2_, the Suzuki-Miyaura coupling reaction could be promoted by the Pd-containing porous polymer.

## 4. Conclusions

Aza-Michael addition reaction of AM4 and HDA in H_2_O successfully yielded porous polymers via polymerization-induced phase separation. The reaction conditions, monomer concentration, reaction temperature and feed molar ratio of AM4 to HDA, strongly affected the surface morphology of the resultant porous polymers. The reactions with low monomer concentrations and at low temperatures tended to yield the porous polymers composed of connected spheres, which could be induced by spinodal decomposition. In contrast, porous polymers with isolated and/connected holes, which were induced by HIPE, were preferentially obtained in the reactions with high monomer concentrations and at high reaction temperatures. The porous polymers were soft and flexible and were not broken by compression. The porous polymer absorbed various solvents, especially protic polar solvents like H_2_O and EtOH, due to the high affinity between the network and the solvents.

The surface of the AM4-HDA porous polymer could be plated with Ni. However, the plating of inside of the porous polymer was impossible because the Ni plated surface hindered further plating. A Pd(acac)_2_-treated porous polymer promoted Suzuki-Miyaura coupling reactions of 4-bromoanisole and phenylboronic acid.

The aza-Michael addition reaction of the multifunctional acrylamide and diamine compounds in H_2_O is one of the effective methods to synthesize the porous polymers, based on the joint and linker concept. As the next step, we are studying the precise control of the morphology of the porous polymers in combination with theoretical simulation studies of the formation process of the network. The fabrication of columns, sheets and discs by the present porous polymer should be useful for further applications of the porous polymers, such as separation columns, supports for conductive materials, battery separators, scaffolds for cell cultivation etc. These investigations are now underway and the results will be reported elsewhere in due course.

## Data Availability

The data presented in this study are available in insert article and Appendix A here.

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
