# Peer review of "Morphology Control and Metallization of Porous Polymers Synthesized by Michael Addition Reactions of a Multi-Functional Acrylamide with a Diamine"

_materials, 2021, doi:10.3390/ma14040800_

Round 1

Reviewer 1 Report

The contribution presents results on application of Aza-Michael addition reaction in the synthesis of porous polymers. The theme of the investigations is up to date but in my opinion manuscript has serious flows. As for details:

  1. The introduction part does not deliver satisfactory insight into the state of the art. Only one paragraph was devoted to this and was written in quite unclear way. What is more, the references (only 19) are rather incidental, very general, not connected with specified applications or synthesis method.
  2. The authors did not give proper codes to the polymers. Instead of it, they use a descriptive system which is rather incomprehensible.
  3. Line 81 HAD is diamine, not “has two primary amines, NH2 group”.
  4. Fig. 2 (a) presents SEM mage of polymers obtained at 70°C not “less than 70°C” as claimed in line 159.
  5. As the obtained polymers are porous, detailed discussion on porous structure should be the part of the work. Instead of this the authors provide only one parameter (surface area) for only one polymer. What is more, Figure S1 presents only nitrogen adsorption/desorption isotherms not BET plot.
  6. The manuscript is written carelessly (e.g. quotation marks after parentheses, distorted unit of the size in Table 1, inconsistent markings system, lots of language mistakes)

In my opinion, this work does not meet the requirements regarding scientific papers and does not make a noticable contribution to the field.  

Author Response

Thank you very much for your letter of Dec. 21, 2020 and for the referees’ comments concerning our manuscript entitled "Morphology control and metallization of porous polymers synthesized by Michael addition reaction of multi-functional acryl amide with diamine " (materials-1025406). We have studied their comments carefully and have made corrections which we hope to meet with their approval. Revised and added portions are written in red.

1.Introduction has been drastically modified to clear novelty of the manuscript. References concerning applications and methods were added to the revised manuscript.

We hope that these meet with the reviewer’s approval.

  1. Code of samples is defined as following, Case of AM4/HDA feed ratio-monomer concentration-temperature.

3.4. The descriptions have been corrected according to suggestion of the reviewer.

  1. The related surface areas of other porous polymers were too low to evaluate quantitatively. The low value of surface area is derived from not the “micro-porous” but “macro-porous” structure. The related surface areas are not suitable to discuss about the surface morphology of the present porous polymers. The macro-porous surface structure of the porous polymers was observed by SEM and analyzed with histograms of diameters of spheres and holes. Precise discussion of surface morphology with schematic explanations of formation process was also added to the revised manuscript.

6.We will have native check before publication.

Reviewer 2 Report

The authors reported the synthesis of a series of porous polymers by aza-Michael addition reaction. Morphology control and solvent adsorption performance of the porous polymers were studied. Initial application of the porous polymers as catalyst carrier was also presented. Overall, the description in this manuscript is not very clear and the novelty is low. Thus I don't think this manuscript is deserved to be published in Materials.

Author Response

Thank you very much for your letter of Dec. 21, 2020 and for the referees’ comments concerning our manuscript entitled "Morphology control and metallization of porous polymers synthesized by Michael addition reaction of multi-functional acryl amide with diamine " (materials-1025406). We have studied their comments carefully and have made corrections which we hope to meet with their approval. Revised and added portions are written in red.

Introduction has been drastically modified to clear novelty of the manuscript. Precise discussion of surface morphology with schematic explanations of formation process was also added to the revised manuscript. We hope the revised manuscript meets with the reviewer’s approval.

Reviewer 3 Report

The reviewed manuscript, number materials-1025406, titled: ‘Morphology control and metallization of porous polymers synthesized by Michael addition reaction of multi-functional acryl amide with diamine’ presents the interesting study of the synthesis, characterization and application of porous crosslinked polymers. The proposed method of the obtaining of porous polymers could be considered as ‘green’ since the reaction is carried out in water environment without using any emulsifiers or stabilizers. The manuscript is well written, in my opinion it is worth to be published in Materials after a minor revision.

  1. Paragraph 2.2, line 86: 0.117g of HDA it is not 0.62 mmol!
  2. What was the form/shape of obtained polymers? Did they possess the shape of the ampoule (block) or maybe they were powders?
  3. Page 6, line 216-219: concerning porous structure, the value of presented surface area is really low compared with traditional porous polymers. From presented data (Fig S1) it is difficult to conclude that discussed polymer is macroporous material. To prove macroporous structure, the pore size distribution diagram should be presented in the manuscript.
  4. Table 1: what does it mean the in the caption ‘morphology’? Is it the morphology of the polymer? And next: ‘size’ – size of what? and what is the unit of size (there in manuscript is written [m])?
  5. Fig S4 in supplementary file – there is no description of points on the diagram; it is difficult to guess which point corresponds to which solvent.
  6. There is lack of Figure S6 in supplementary file.
  7. Concerning coupling reaction (Paragraph 3.4), it does not appear from the presented discussion that the Suzuki-Miyaura reaction was promoted by the Pd-porous polymer. Authors should give the yield of such reaction carried out without the polymer and also with the use of another Pd-contained catalyst to compare obtained results. The sentence ‘The reaction did not progress in EtOH/H2O solvent, which was used for washing of the Pd catalyzed AM4-HDA porous polymer’ is incomprehensible. What was the form of polymer added to the reaction mixture?
  8. Page 11, line 358: Authors have written: ‘High tolerance to hydrolytic degradation of acrylamide group in the AM4-HDA porous polymers’, I wonder if Authors study the chemical structure of obtained polymers after plating process? Why they assumed that it was unchanged after the plating? And generally, did Authors study the chemical structure of prepared polymers?

Author Response

Thank you very much for your letter of Dec. 21, 2020 and for the referees’ comments concerning our manuscript entitled "Morphology control and metallization of porous polymers synthesized by Michael addition reaction of multi-functional acryl amide with diamine " (materials-1025406). We have studied their comments carefully and have made corrections which we hope to meet with their approval. Revised and added portions are written in red.

  1. The pointed value has been corrected.
  2. A picture of a sample was added in Figure 4.

3.We don’t have mercury porosimeter to analyze pore size distribution. The macro-porous surface structure of the porous polymers was quantitatively analyzed with histograms of diameters of spheres and holes.

4.morphology à surface morphology, size -> diameter (of spheres or holes), m -> micro

5.Names of the solvents were added in the Figure.

6.We concluded that Figure S6 of the original manuscript is not necessary. The Figure was deleted in the revised manuscript.

  1. A Pd(acac)2 catalyzed AM4-HDA porous polymer was cut to small blocks about 1 mm, then was added into the flask and stirred at 55 °C for 3 h.

The conversion of the corresponding reaction with Pd(acac)2 in homogeneous system was more than 98 %. The reaction did not progress with the original AM4-HDA porous polymer.

These explanations have been added in the revised manuscript.

We don’t have another Pd supported catalyst now.

8.The chemical structure of the catalyzed porous polymer was studied by FT-IR spectroscopy. But the profile was complicated to discuss about the stability of the polymer network due to the existence of Pd(acac)2. If the hydrolytic degradation occurred, the porous structure could not be kept not only in the plating (catalyzation) process but polymerization (preparation) process.

Reviewer 4 Report

See attached file

Author Response

Thank you very much for your letter of Dec. 21, 2020 and for the referees’ comments concerning our manuscript entitled "Morphology control and metallization of porous polymers synthesized by Michael addition reaction of multi-functional acryl amide with diamine " (materials-1025406). We have studied their comments carefully and have made corrections which we hope to meet with their approval. Revised and added portions are written in red.

  1. The reaction can accelerate by a catalyst. But fast reaction rate is not always good to obtain porous polymers. When the polymerization (network formation) rate is higher than the phase separation rate, the reaction tends to yield precipitates, as previously reported in other reaction systems. In the present study, the reactions without catalyst were suitable to obtain the porous polymers.
  2. All porous polymers are not thermoset materials.
  3. DSC measurement of a porous polymer was added in the revied manuscript. The profile showed grass transition and thermal degradation.
  4. Figures 2 and 3 have been corrected accordingly.
  5. A picture of a sample was added in Figure 4.

Reviewer 5 Report

This manuscript concerns the synthesis and characterisation of porous polymers. The paper, figures and information is generally well-presented. However, I feel there are some refinements that could help to clarify and improve the paper, prior to its acceptance for publication. 

  • Perhaps a clear statement of the experimental hypothesis could be clearly stated at the conclusion of the introduction in order to make explicitly clear, the aims of the project.
  • I found the distinction between spheres and holes etc., hard to follow. Perhaps a clear schematic of the process and the differences between the various samples would allow for a clearer understanding of the results as well as a better appreciation of the data.
  • Important methodological details are missing. For example;
    • NMR: what solvent and reference standard was used?
    • SEM: Were samples coated prior to analysis? How? With what material?
  • The number of references seems a little low, especially for an area and topic with a rich and contemporary amount of overlapping and relevant literature. 

Author Response

Thank you very much for your letter of Dec. 21, 2020 and for the referees’ comments concerning our manuscript entitled "Morphology control and metallization of porous polymers synthesized by Michael addition reaction of multi-functional acryl amide with diamine " (materials-1025406). We have studied their comments carefully and have made corrections which we hope to meet with their approval. Revised and added portions are written in red.

1.We have drastically modified introduction which we hope to meet with their approval.

2.A model scheme of the formation process of the porous polymer with surface structures (spheres and holes) has been added in the revised manuscript.

3.Experimetal details are added in the revised manuscript for the benefit of readers.

4.References concerning applications and methods were added to the revised manuscript.

Round 2

Reviewer 1 Report

The revised manuscript entitled “Morphology control and metallization of porous polymers synthesized by Michael addition reaction of multi-functional acryl amide with diamine” contains new interesting information but in my opinion there are still deficiencies. As for details:

  1. In the Introduction part examples of important application of porous polymers were added but without adequate references.
  2. Figure S1 contain only nitrogen adsorption-desorption isotherms. The caption should be corrected.
  3. DSC curve analysis is not done correctly. The shape of the endothermic peak at around 90 °C is typical for water desorption. It certainly cannot be related with the glass transition.
  4. The authors claimed ”We will have native check before publication”. I think that a good solution is to have “check” before submission.

In my opinion, this work can be published in Materials after major revision.

Author Response

1.The references of applications of porous polymers were added in the re-revised manuscript.

  1. The caption of Figure S1 was corrected accordingly.
  2. The explanation of the DSC profile was corrected accordingly.
  3. Native check of the manuscript is in progress. We will have the check before submission from next time.

Reviewer 2 Report

The revised manuscript doesn't show reasonable improvement.

Author Response

We have re-revised the manuscript according to the comments of other referees. We hope the re-revised manuscript meets with the reviewer’s approval.

Reviewer 3 Report

The new revision of the manuscript is better than the previous one. However, some mistakes are still present in the text and need to be corrected.

The discussion of DSC results is incorrect. The endothermic peak with max at about 90oC is not derived from glass transition for sure! It looks like the peak connected with the evaporation of solvent, most probably water.

In Fig S1, the description of curves is incorrect.

Author Response

1.The explanation of the DSC profile was corrected accordingly.

  1. The caption of Figure S1 was corrected accordingly.

Reviewer 4 Report

The comments have been addressed in the revised version of the manuscript.

Author Response

We have re-revised the manuscripts which we hope to meet with the referee's approval.

Reviewer 5 Report

This experimental article resubmission is much improved from the prior submission. The details, justification and organisation are significantly improved.

My only concern is, in certain parts, the quality of writing, which is often difficult to follow. For example, the conclusions section is extremely difficult to understand. I would recommend re-writing this entire section.

With the above changes and a thorough readthrough for readability, the paper should be suitable for eventual publication.

Author Response

  1. Native check of the manuscript is in progress. We will have the check before submission from next time.